# Patient safety in a rural sub-Saharan Africa hospital: A 7-year experience at the AIC Kijabe Hospital, Kenya

**Peter M. Nthumba** [1,2]*, **Caroline Mwangi**[1], **Moses Odhiambo**[1]

**1** Research Department, AIC Kijabe Hospital, Kijabe, Kenya, **2** Department of Plastic Surgery, Vanderbilt University Medical College, Nashville, Tennessee, United States of America

* nthumba@gmail.com

## Abstract

The development of a safety culture is challenging, primarily because it often disrupts institutional attitudes, norms and values. In the healthcare industry, most of the data on the results of unsafe care come from High-Income Countries. The Hospital Survey on Patient Safety Culture (HSOPS) is a tool for assessing, building, sustaining and comparing institutional safety cultures within healthcare organizations. We used the HSOPS over a 7-year period, and herein report our experience. The authors report their experience using the HSOPS tool in Kijabe Hospital, an institution with 650 employees, over a 7-year period. The HSOPS tool, with no local modifications, was distributed to all employees during each survey. The institutional HSOPS percent positive dimension scores for 2015, 2017 and 2019 were compared with baseline data from the 2013 survey. The average response rate during the study period was 84.5% (range 65.1% to 93.6%). In general, the mean percentage positive dimension scores of most domains improved in the 2019 survey (p<0.05), including reduced staff turnover and, improved hospital support for patient safety (p<0.0001), amongst other domains. The overall patient safety grade (excellent/very good), was 50% (range 43–64%). Although the dynamics of high staff turnover and hospital leadership change presented challenges in developing and measuring institutional patient safety culture, this study demonstrates that patient safety ideals can be developed and embraced in sub-Saharan Africa. Patient safety champions, a generative institutional leadership that is supportive of patient safety, are important for the development of an institutional safety culture. Creating an institutional just culture creates a patient safety culture.

## Introduction

Safety culture has been defined as "the extent to which individuals and groups will commit to personal responsibility for safety, act to preserve, enhance and communicate safety concerns, strive to actively learn, adapt and modify behavior based on lessons learned from mistakes, and be rewarded in a manner consistent with these values" [1, 2]. The performance of an organization is a reflection of its culture: its shared attitudes, values, norms, underlying

**Data Availability Statement:** Summated data are attached as supporting information.

**Funding:** The authors received no specific funding for this work.

**Competing interests:** The authors have declared that no competing interests exist.

assumptions and behaviors, or "the way we do things around here" [3]. The development of a safety culture is challenging, primarily because it often disrupts the beliefs and practices held dear by those within the organization, which upon scrutiny may be found to 'enable or condition the opportunities for active error'. Efforts to improve patient safety should therefore focus on causal relationships between system factors and individual staff error [4]. The Institute of Medicine in its seminal publication, "*To Err is Human*" stated that "*the biggest challenge to moving toward a safer health system is changing the culture from one of blaming individuals for errors to one in which errors are treated not as personal failures, but as opportunities to improve the system and prevent harm*" [5].

Patient safety has been defined as "the avoidance and prevention of patient injuries or adverse events resulting from the processes of healthcare delivery." [6]. A heightened awareness of patient safety in healthcare institutions helps ensure safe transit for patients during their healthcare encounter.

Most of the evidence on the outcomes of unsafe care comes from High-Income Countries (HICs), where between 3% and 16% of hospitalized patients suffer harm during medical care [7]. Furthermore, medical error is estimated to cause 30% of deaths in the United States of America healthcare system [8]. A model for reducing patient harm from individual and system errors in healthcare in which errors are made more visible, responded to, and steps taken to make errors less frequent has been proposed to counter these grim statistics [8, 9].

The World Health Organization (WHO) estimates that up to 134 million adverse events occur each year due to unsafe care in hospitals in low-and-middle-income countries (LMICs), contributing to 2.6 million deaths annually [10]. Wilson et al. reported an adverse event rate of 8.2% across eight LMICs, and 83% of the adverse events were preventable. Furthermore, these authors established that failures in clinical process rather than the absence of essential resources accounted for most contributory factors to the adverse events [11].

Jha et al. identified 23 major patient safety topics from different types and causes of adverse events that are particularly harmful to patients. Seven topics that are especially relevant to the patient safety agenda in LMICs include: safety culture; stress and fatigue; inadequate training, education, and manpower issues; lack of appropriate knowledge and availability of knowledge, transfer of knowledge; errors in the structure and process of care; adverse events due to healthcare associated infections; and 'How to bring the patients' voices into the patient safety agenda.' [7].

Patient safety as a substantive government agenda in Africa may have first been discussed in January 2005 in meetings held in Nairobi, Kenya, and Durban, South Africa; the meetings emphasized the need for quality and safer care through medical error prevention. In 2007, Ministry of Health officials from twenty African countries gathered in Rwanda for the 'First awareness raising workshop on patient safety issues in the African region' [12]; subsequently, the WHO Africa Regional Director wrote a report on the status of patient safety in Africa in 2018 [13]. In September 2019, countries across Africa marked the first ever World Patient Safety Day, a day set aside by world leaders at the 72nd World Health Assembly to create awareness on patient safety and commitment towards making healthcare safer. The WHO Africa Regional Director noted the following barriers to the realization of a robust patient safety climate in Africa: 'a lack of national policies, strategies, standards, guidelines and tools for safe healthcare practices, ineffective implementation where they exist, inadequate funding; inadequate human resources for health, weak healthcare delivery systems with suboptimal infrastructure, poor management capacity and under-equipped health facilities; and ineffective mechanisms for forging strong partnerships to adequately involve patients and civil society in the improvement of patient safety' [14, 15]. Konlan et al. conducted a comprehensive review of factors that influence patient safety in Africa, and their findings reflect those stated by the

Regional WHO Africa director [16]. Aveling et al., in a study titled 'Why is patient safety so hard in LMICS?' concluded that the appropriate structures of governance, management and accountability must be in place for patient safety to grow in Africa. Nevertheless, these authors noted that irrespective of the resource environment, the obstacles and solutions to patient safety problems are "rooted in human factors, resources, culture and behavior" [17]. Ente et al. found that HCWs in Africa were aware of medical errors, experienced errors and were willing to play active roles in reducing and preventing such errors, and concluded that no meaningful progress in patient safety would occur without the active involvement of governments [18].

The WHO helped create African Partnerships for Patient Safety, which focused on building strong patient safety partnerships between hospitals in Africa and Europe [19]. A number of other projects with the goal of entrenching a strong patient safety culture in African healthcare institutions have been implemented; these have helped transfer this agenda from the bureaucracy of government boardrooms to real-life frontlines, providing much needed evidence that patient safety can grow within the continent [20, 21]. Additionally, in some African countries, indigenous efforts have promoted a patient safety agenda [22, 23]. Although some African countries have published patient safety guidelines, the extent to which they are translated into practice is uncertain [24, 25]. Unfortunately, information on patient safety from the region remains, "infrequent and limited in scope" [26]. The limited data from LMICs [11], particularly sub-Sahara Africa [18, 27], the statement made by the US Institute of Medicine: "*One of the most important barriers to increasing patient safety is a lack of awareness of the extent to which errors occur daily in all healthcare settings and organizations.*" [5], and Pronovost's contention that for an institution to improve patient safety, it must assess its patient safety culture [28], all make an urgent case for practical solutions to obstacles preventing the widespread adoption of the patient safety agenda in sub-Saharan Africa.

To create a standardized, sustainable, and replicable system for assessing the patient safety culture across different healthcare institutions, regulators in HICs adopted patient safety survey tools. These tools are used to assess, build, sustain and compare institutional safety cultures within healthcare organizations. The Hospital Survey on Patient Safety Culture (HSOPS) is one of the most widely used [29]. HSOPS has been validated in multiple languages and countries [2, 30–36]. HSOPS highlights patient safety, error and event reporting using 42 items grouped into 12 composite measures. In a recent systematic review of the psychometric properties of international studies that have used the HSOPS tool, Waterson et al. found 62 studies and 67 datasets; they noted a growing worldwide trend in the use of the HOSPS tool. However, they cautioned about the need for guidelines for the adaptation, translation and reporting of results from use of the tool [36]. The use of HSOPS in Africa has been reported in a few studies [37–40]. Mallouli et al. used a French tool to assess patient safety in primary healthcare [41].

Even with the noted limitations, the HSOPS is a useful tool, as it provides a structured, relevant data collection tool for most environments, without the need for extensive adaptation. Although HICs and LMICs are at different levels of engagement within the patient safety agenda, the goals of patient safety culture assessment are nevertheless very similar: increasing awareness of patient safety concepts, identifying areas of culture in need of improvement, evaluating the effectiveness of patient safety interventions over time, and comparing patient safety standards against other hospitals, among other goals [42, 43].

The HSOPS tool was introduced in the authors' hospital (Kijabe Hospital) in 2013 at the beginning of the WHO African Surgical Unit Safety Program (SUSP) project [20], and surveys were repeated every two years thereafter. The authors used the HSOPS tool to establish the baseline status of patient safety culture, raise staff awareness about patient safety and thereafter, examine trends in patient safety culture change over time, and evaluate the institutional cultural impact of patient safety initiatives and interventions [6]. In this report, the authors

describe their institutional patient safety trends over a seven-year period, through a review of HSOPS results, and hope that their experiences will stimulate regional healthcare institutions to start a similar journey.

## Methods

### Ethics statement

This institutional quality improvement project was approved as part of the Surgical Unit Safety Program by the Institutional Ethical Review Board and has been running since 2013. In order to maintain anonymity, a formal consent for participation was not obtained.

The authors reviewed the KH institutional patient safety experiences through longitudinal use of the HSOPS tool. KH is a 360-bed rural teaching hospital with approximately 650 employees. A SUSP core team comprising of six patient safety champions ran the day-to-day activities of the WHO African SUSP project [20]. As part of the baseline SUSP study, the HSOPS tool was administered to all hospital employees in their departments after an explanation of the tool and the data collection process. The HSOPS tool was used without any local modifications. For each survey, the HSOPS tool was printed and given to departmental heads to distribute among team members; it was filled anonymously, with no participant identifiers. In the week before the questionnaire was distributed, the SUSP core team met staff in their departments and taught them about the HSOPS tool, during which the staff were taught the precepts of patient safety and shown the African SUSP patient safety video [44]. Because the total staff were 650 throughout this period, the number of staff reporting having worked 'less than 1 year', were newly employed within the previous 12 months; an increase in the percentage of staff who had worked 'less than 1 year', indicated an increase in the number of new staff, an indication of increased staff turnover. Conversely, a reduction in this category, indicated staff transitioning within the institution, to the '1 to 5 years' category, a surrogate indicator for reduced staff turnover.

In 2013 after the HSOPS analysis, the SUSP core team determined that for feedback purposes, the team would focus on those parts of the tool they felt would help the institution best evaluate and develop a sustainable patient safety culture. The team settled on the domains listed in Table 1 [29].

We performed mixed analysis, (qualitative and quantitative). The HSOPS percentage positive dimension scores for the years 2015, 2017 and 2019 were compared with baseline data (2013).

The percent positive dimension scores were calculated from both positively–and negatively–worded items; the total percent scores were then averaged within each dimension by the number of items to give the mean percentage. Positively-worded responses included the following answers: "Strongly agree" or "Agree," or "Always" or "Most of the time". Positive item responses for negatively-worded items such as 'Staff feel like their mistakes are held against them' included: "Strongly disagree" or "Disagree," or "Never" or "Rarely".

The free comments from the 2019 HSOPS, which were evaluated by five members of the study, were contributed by only 50 participants and analyzed using Nvivo12 (QSR International Pty Ltd., Chadstone, Australia). The team coded the participants free-text comments into broad categories as themes emerged. The final thematic analysis involved merging similar themes into broader categories. Thematic saturation was not anticipated because of the small number of participants who wrote free-text comments. Quantitative analysis was performed using STATA version 18 (StataCorp LLC, College Station, TX, USA), and reported as descriptive statistics. Differences in the proportions of the different variables were used to evaluate changes over time, and reported with P values (a P value of <0.05 was considered statistically significant).

**Table 1. Questions analyzed to determine patient safety awareness and growth over three time periods.**

| Section | | |
|---|---|---|
| A | **Your Work Area/Unit** | |
| | | What is your primary work area or unit in this hospital? |
| | | We have enough staff to handle the workload |
| | | Staff in this unit work longer hours than is best for patient care |
| | | We are actively doing things to improve patient safety |
| | | Staff feel like their mistakes are held against them |
| | | Mistakes have led to positive changes here |
| | | We have patient safety problems in this unit |
| B | **Your Supervisor/ Manager** | |
| | | My supervisor/manager seriously considers staff suggestions for improving patient safety |
| | | In this unit, we discuss ways to prevent errors from happening again |
| C | **Communications** | |
| D | **Frequency of Events Reported** | |
| | | When a mistake is made, but is caught and corrected before affecting the patient, how often is this reported? |
| | | When a mistake is made that could harm the patient, but does not, how often is this reported? |
| E | **Patient Safety Grade** | |
| | | Please give your work area/unit in this hospital an overall grade on patient safety. |
| F | **Your Hospital** | |
| | | Hospital management provides a work climate that promotes patient safety |
| G | **Number of Events Reported** | |
| | | In the past 12 months, how many event reports have you filled out and submitted? |
| H | **Background Information** | |
| | | How long have you worked in this hospital? |
| | | How long have you worked in your current hospital work area/unit? |
| | | Typically, how many hours per week do you work in this hospital? |
| | | What is your staff position in this hospital? Select ONE answer that best describes your staff position. |
| | | In your staff position, do you typically have direct interaction or contact with patients? |
| | | How long have you worked in your current specialty or profession? |
| I | **Your Comments** | |

Response rates were calculated using the formula: $\frac{Number\ of\ surveys\ returned\ -incompletes}{Number\ of\ eligible\ staff\ who\ received\ a\ survey}$

The authors acquired data on staff numbers, indicating number of new staff hired, and those that had left KH during the 12 months prior to administration of the HSOPS.

All the HSOPS results were reported to, and discussed with senior management; the senior management effected changes to address some of the issues raised in each report.

## Results

The results were tabulated and the mean percent positive dimension scores for each year were calculated (Table 2). Table 3 shows participant demographics and responses to some questions

**Table 2. HSOPS mean percent positive responses for years 2013, 2015, 2017 and 2019.**

| | Mean percent positive dimension scores | | | | Difference from baseline | | | | | | | | |
|---|---|---|---|---|---|---|---|---|---|---|---|---|---|
| | 2013 | 2015 | 2017 | 2019 | 2015–2013 | $\pi^2$ | p value | 2017–2013 | $\pi^2$ | p value | 2019–2013 | $\pi^2$ | P value |
| **Response rates N (%)** | 350 (87.5) | 307 (90.3) | 281 (93.6) | 241 (65.1) | a 2.8 | a 1.29 | 0.256 | b 6.1 | b 6.56 | 0.01* | c -22.4 | c 42.16 | <0.0001* |
| **Outcomes** | | | | | | | | | | | | | |
| Overall perceptions of patient safety | 41 | 41.6 | 36 | 42.6 | 0.6 | 0.024 | 0.88 | -5 | 1.64 | 0.20 | 1.6 | 0.15 | 0.70 |
| Patient safety grade (excellent/very good) | 50 | 56 | 64 | 43.4 | 6 | 2.35 | 0.12 | 14 | 12.39 | 0.0004* | -6.6 | 2.49 | 0.11 |
| Frequency of event reporting | 53 | 68.4 | 65 | 61.5 | 15.4 | 16.16 | 0.0001* | 12 | 9.22 | 0.002* | 8.5 | 4.19 | 0.04* |
| **Hospital composite dimensions** | | | | | | | | | | | | | |
| Hospital management support for patient safety | 62.9 | 62.7 | 65 | 80.8 | -0.2 | 0.003 | 0.96 | 2.1 | 0.297 | 0.59 | 17.9 | 21.82 | <0.0001* |
| **Unit-composite dimensions** | | | | | | | | | | | | | |
| Staff work longer hours than is best for patient | 46 | 46.3 | 43 | 29.9 | 0.3 | 0.006 | 0.939 | -3 | 0.56 | 0.45 | 16.1 | 15.46 | 0.0001* |
| Patient safety problems in this unit | 36 | 40 | 49 | 30.5 | 4 | 1.11 | 0.29 | 13 | 10.81 | 0.001* | -5.5 | 1.93 | 0.165 |
| Mistakes have led to positive changes | 50 | 62 | 54 | 68.8 | 12 | 9.53 | 0.002* | 4 | 0.91 | 0.33 | 18.8 | 20.62 | <0.0001* |
| Staff actively improving patient safety | 52 | 60.3 | 53 | 92.9 | 8.3 | 4.56 | 0.03* | 1 | 0.06 | 0.80 | 40.9 | 110.8 | <0.0001* |
| Supervisor considers staff suggestions | 51 | 56 | 55 | 74.2 | 5 | 1.64 | 0.20 | 4 | 0.99 | 0.318 | 23.2 | 32.48 | <0.0001* |
| Staffing | 35 | 41 | 45 | 37.2 | 6 | 2.5 | 0.114 | 10 | 6.51 | 0.011* | 2.2 | 0.30 | 0.584 |
| Non-punitive response to errors | 29 | 20 | 28 | 27 | -9 | 7.09 | 0.0077* | -1 | 0.076 | 0.782 | -2 | 0.28 | 0.596 |
| Feedback and communication | 44 | 44.6 | 46 | 36 | 0.6 | 0.02 | 0.877 | 2 | 0.25 | 0.62 | -8 | 3.77 | 0.052 |

* denotes P-value <0.05, indicating a statistically significant difference from the baseline value

on patient safety. Chi-square tests were used to analyze the differences in proportions of mean percent positive dimension scores between each year and the baseline HSOPS scores (Table 2).

## Descriptive statistics

The average response rate was 84.5% (65.1%-93.6%), with between 241 and 350 respondents each year (Table 2). The only missing data from the surveys were the number of hours worked in 2013. In general, however, the majority of staff (80% to 87.2%) worked between 40 and 49 hours.

Because the missing data affected a single category in 2013 (working hours per week), it did not affect data analysis; we did not perform any statistical comparisons in this category. The scores reported in 2013, 2015 and 2017 were similar, with the largest changes noted in 2019.

## Respondents position in the hospital

Hospital staff surveyed included nursing staff, clinical officers, physicians, pharmacy staff, ward clerks, and others (staff not generally in direct contact with patients during normal work, including Infection Prevention and Control staff, medical records, housekeeping, and security personnel).

## Years worked

On average, 27.9% and 23.2% of staff had worked in the hospital for one year, and one to five years respectively. The number of staff within the bracket of '<1 year worked' was used as a surrogate measure of staff turnover. Because the total staff population was fairly constant, and

**Table 3. HSOPS staff responses for years 2013, 2015, 2017 and 2019.**

| Question | Response | 2013 (350) | 2015 (N = 307) | 2017 (N = 281) | 2019 (N = 241) |
|---|---|---|---|---|---|
| Respondents position in the hospital | Registered nurse | 44.0 | 56.0 | 60.0 | 50.0 |
| | Clinical Officers | 2.0 | 3.1 | 3.0 | 7.5 |
| | Doctors | 11.0 | 4.3 | 8.0 | 6.3 |
| | Pharmacist | 2.0 | 8.0 | 5.0 | 5.9 |
| | Ward secretary | 1.0 | 0.6 | 2.0 | 0.8 |
| | others | 40.0 | 28.0 | 22.0 | 29.5 |
| Years worked in hospital | Less than 1 Year | 20.6 | 38.8 | 35.0 | 17.1 |
| | 1 to 5 Years | 53.4 | 38.8 | 30.0 | 50.6 |
| | 6 to 10 Years | 11.1 | 12.4 | 10.0 | 14.1 |
| | 11 to 15 Years | 8.3 | 4.9 | 8.0 | 9.1 |
| | 16 to 20 Years | 3.7 | 3.1 | 10.0 | 3.7 |
| | 21 years or more | 2.9 | 2.0 | 7.0 | 5.4 |
| Total hospital staff turnover | | | **16%** | **21%** | **10%** |
| Working hours per week | Less than 20 hours | Missing | 2.0 | 4.0 | 2.1 |
| | 20 to 39 hours | | 3.3 | 5.0 | 3.9 |
| | 40 to 59 hours | | 87.2 | 80.0 | 83.7 |
| | 60 to 79 hours | | 2.3 | 4.0 | 7.3 |
| | 80 to 99 hours | | 3.9 | 5.0 | 2.6 |
| | 100 or more hours | | 1.3 | 2.0 | 0.4 |
| Mistakes caught and corrected before affecting patients | Always | 24.0 | 26.0 | 30.0 | 13.9 |
| | Most of the time | 26.0 | 30.0 | 34.0 | 29.5 |
| | Sometime | 25.0 | 30.0 | 20.0 | 30.0 |
| | Rarely | 20.0 | 9.0 | 8.0 | 22.8 |
| | Never | 5.0 | 5.0 | 8.0 | 3.8 |
| We discuss ways to prevent errors from happening again | Always | 1.8 | 29.0 | 12.0 | 1.7 |
| | Most of the time | 33.6 | 33.0 | 10.0 | 5.0 |
| | Sometime | 26.6 | 24.4 | 22.0 | 25.3 |
| | Rarely | 28.0 | 8.7 | 30.0 | 33.6 |
| | Never | 1.8 | 5.2 | 26.0 | 34.4 |
| How many events reports have you filled? | No events | 53.0 | 68.0 | 65.0 | 61.5 |
| | 1–2 events | 22.0 | 16.0 | 10.0 | 25.5 |
| | 3–5 events | 18.0 | 7.5 | 16.0 | 4.8 |
| | 6–10 events | 4.3 | 4.6 | 3.0 | 2.9 |
| | 11–20 events | 1.4 | 1.9 | 2.0 | 3.9 |
| | ≥21 events | 1.3 | 1.6 | 4.0 | 1.4 |

the sample surveyed fairly representative, an increase in staff numbers '<1 year worked' indicated replacement of staff that had transitioned to other institutions, while a decrease indicated staff transitioning to the '1 to 5 years' bracket, hence a potential reduction in staff turnover. There was steady growth in the number of staff who had worked for a year or less, in 2015 and 2017, compared to 2013, $p<0.001$); this trend ended in 2019, with a return to 2013 levels. The increases in 2015 and 2017 in staff in the '<1 year' category was mirrored by decreases in the 'year 1 to 5' category over the same period ($p<0.001$) (Table 3). The increased staff turnover correlated well with the overall institutional turnover reported by the Human Resources department for the period under study. It is noteworthy that about 50% of hospital employees are nurses, the main drivers of turnover. The staff turnover was calculated during the year covered by the two-yearly HSOPS.

## Patient safety

Hospital management support for patient safety was much higher in 2019 than in 2013 (p<0.0001), at 80.8%, and in the same year, the perception that staff hours did not potentially endanger patients fell to 29.9%; p = 0.0001. Additionally, mistakes led to positive changes in patient safety with statistically significant peaks in 2015 and 2019. Except for a dip in 2015, the percentage of staff who felt that the administration's response to errors was nonpunitive remained the same. In general, most domains improved in the 2019 survey mean percent positive dimension scores (p<0.05) (Table 2).

## Qualitative analysis

Eleven major themes emerged from the analysis of the free comments from the 2019 survey (Table 4 and Fig 1). Problems with equipment constituted 21.3% of the staff concerns, while 20.7% of staff were appreciative of the administrative support for patient safety. Whereas the free text comments that constituted the qualitative data included important thematic areas affecting patient safety, the fact that only a small proportion of participants included comments in their survey forms may represent a selection bias. For example, the comments may have come primarily from departmental managers and senior staff.

## Institutional changes

Beginning in 2013, after the first HSOPS, the hospital administration created the 'Infection Prevention and Control (IPC) Committee'. Departmental IPC champions were appointed to the committee to spearhead IPC and patient safety issues within their departments. An important member of the SUSP team was the 'executive member'; the human resources director, a senior member of the executive, performed this role. The SUSP core team conducted monthly hospital tours to determine patient safety issues on the ground, as well as infrastructural impediments to a safe environment. This representation and support at the highest decision-making organ of the hospital, and the 'open-door' policy for reporting issues directly, helped galvanize institutional support, with appropriate resources for improving patient safety.

The hospital employed a second full-time IPC nurse. The SUSP core team reported monthly to the IPC Committee on all ongoing patient safety and IPC issues, and also discussed the results of each HSOPS. Departmental representatives championed departmental-level initiatives, while the IPC chair reported directly to the hospital administration. As a result of these activities, water heaters were installed in patient bathrooms, and ward and bathroom walkways were cleared of cluttering which included clutches and broken-down equipment, enabling patients to walk safely to bathrooms and take warm showers. Patient feedback was encouraged and letter boxes were strategically placed to encourage patient and relative feedback. Additionally, among many other implemented change the use of the modified WHO checklist was enforced in the operating room for every procedure.

The changes implemented after 2019 were the result of this long journey of incremental change as the institutional safety climate grew. A Quality Improvement (QI) department was created to sustainably foster patient safety, IPC, and other QI initiatives within the hospital. This department is headed by a doctor, with staff including a quality officer and IPC nurses, and is supported by the research office with data analysis and reports. These reports are fed back to clinicians and the administration. The department initiated the process of getting the hospital accredited by SafeCare (https://www.safe-care.org/who-we-are/safecare-standards/. Accessed September 7th, 2024), a journey marked by initiatives to improve the quality of care provided by the hospital. This journey can be traced to HSOPS-results initiatives, reflecting their incremental and impactful nature. An additional, but important offshoot of the safety

**Table 4. Major themes arising from the free comments section.**

| Major themes | Discussion points | Changes effected |
|---|---|---|
| Equipment | • Hospital beds lack rails and proper patient restraints; wheelchairs, stretchers, trolleys, and exam couches need repair<br>• X-ray lead gowns are needed for patient protection.<br>• Provide back friendly seats; warm bathing water, and heaters for very cold wards/rooms to ensure patients are comfortable at night. | |
| Culture change | • Staff have become more responsible after culture change training<br>• Management has provided good climate for patient safety; patient safety is a priority at all times<br>• Medical error data collection tool adopted has helped improve patient safety in pharmacy | |
| Customer care | • There should be someone to direct patients and relatives to different departments within the hospital.<br>• Waiting areas are congested, waiting time is too long, *paperwork should be done earlier before patient taken to theatre*.<br>• *Washrooms are inadequate*; treatment rooms at physiotherapy department are very small and poorly ventilated for patients and care givers<br>• Uncovered pathway towards private clinic/palliative makes patient very vulnerable when it's raining, slippery floors should be changed, especially in hallways | • Electronic Medical Registry (paperless) was launched in September of 2019, enabling efficient documentation and reduction of loss of documentation.<br>• The patient washrooms were renovated to increase capacity and enhance privacy |
| Patient safety concerns | • Have drill to prepare all staff and patient on patient safety<br>• Emergency exit including stairs in private ward should be improved<br>• Need for better system for handing over and picking critical issues on our patient. | |
| Staff safety concerns | • Fire exit is needed at physiotherapy department<br>• Provide adjustable beds with rails for patients at BKKH some are too low hence back pains to staffs | |
| Culture of reporting incidents | • *Encourage culture of reporting incidents even minor ones; review to learn from them.*<br>• *Adopt honest medical legal policy to tell the truth about any error when it occurs to any patient.*• Frequent Customer Due Diligence with regards to patient safety is recommended for recognition, documentation, prevention and prompt measures to taken. | • A hospital-wide audit is held every Friday. Each Friday, a different department presents its audit, as well as any quality improvement projects.<br>• As a policy, the hospital staff initiate early family conferences with full disclosure on any adverse events or errors noted. These discussions are documented in the patient records as well. |
| Blame culture | • Management should avoid blame game when errors occur.<br>• When an error occurs, it should be reviewed so that we can learn from it. | |
| Workplace interaction | • Respect between staff and managers should be improved<br>• Co-operation between departments should be improved<br>• There is need for better system for handing over and picking critical issues on our patient (Handing over SOPs) | |
| Infection control | • Equip security officers with modern security equipment e.g. door scanners to reduce body contact<br>• There is uncovered and stinking drainage under Customer Care office | |
| Inclusivity | • Involve care givers on the ground before putting patient safety measures in place<br>• There is need for consultation on materials to be bought for use by care givers | |
| Information sharing | • *Communication systems should be improved*<br>• *Prompt survey feed back*• Make available standard policy documents at all times. | • A number of WhatsApp groups were created to meet this need amongst staff. Hospital signage was renovated, to provide clear directions to patients and relatives.<br>• A corporate affairs department was created to oversee patient/family–hospital communications (including patient feedback).<br>• Any Quality Improvement surveys amongst staff are reported back within a month |

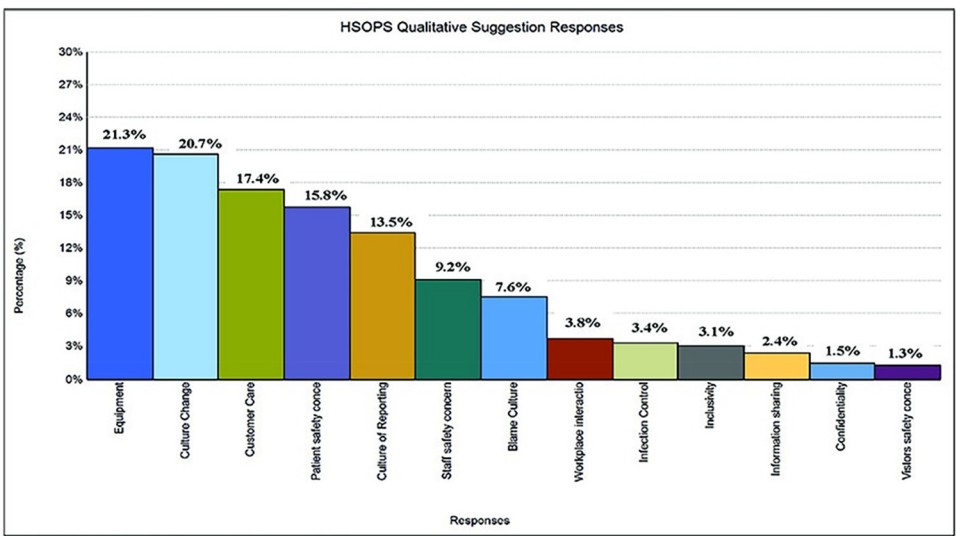

**Fig 1. Analysis of free comments on the HSOPS responses.**

culture journey was the introduction of initiatives to improve the staff working environment, including mental health support systems. This journey is illustrated in Fig 2.

## Discussion

HSOPS was introduced at the AIC Kijabe Hospital in 2013; the overall patient safety grade (excellent/very good), was 50% (range 43% to 64%), a low but acceptable patient safety grade. [6] The number of staff members not filing an incident report increased from the baseline of 53% in 2013; the increase for each year analyzed was higher than the baseline (p<0.05) (Table 3).

In comparison to similar surveys from neighboring Ethiopia, patient safety perception (between 44% and 46%) was higher in our institution [37–39]. The lowest survey response rate

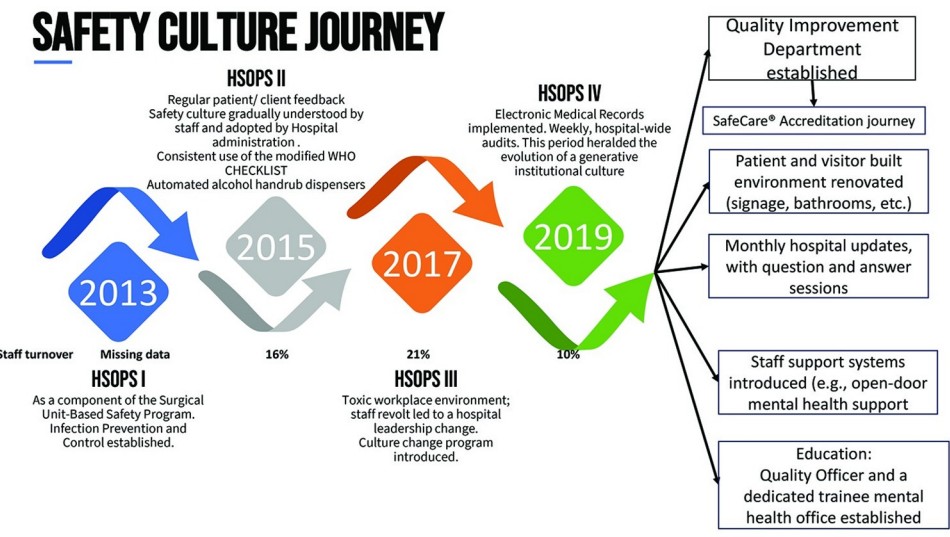

**Fig 2. Institutional changes implemented along safety culture journey.**

was 65.1% in 2019, which was much lower than previous response rates (p<0.05). This was nevertheless above the proposed minimum of 50% response rate by the Agency for Healthcare Research and Quality (AHRQ) [6].

The initial survey helped to establish the concepts of patient safety culture, and raise staff awareness about patient safety. Subsequent HSOPS permitted an examination of institutional trends in patient safety culture over time. Three dominant types of institutional culture, largely shaped by leadership, have been described: pathological (defined by a punitive environment in which the focus is on power, needs, and glory); bureaucratic (information is used to ensure adherence to rules, positions, and to protect turf), and generative (leadership is institutional mission oriented) [45]. Westrum surmised that the climate created by these cultures shapes activities such as communication, cooperation, innovation, and problem solving; he theorized that organizational culture bears a predictive relationship with safety and that particular kinds of organizational culture improve safety [45].

KH has traditionally had a high staff turnover (a situation not peculiar to KH) [40]. Because hospital administrators and physicians often perceive their institution as having a better patient safety culture than frontline employees [46, 47], the KH leadership perception of staff satisfaction with their working conditions, as well as the explanation for the high staff turnover, was misleading.

Prior to 2015, because of the structure of the KH nursing program, all graduates from the program were immediately absorbed into the labor force, either at KH or some other KH-affiliated healthcare institutions (including rural health centers and dispensaries), based on contracts between graduands and training-sponsoring institutions. The KH leadership determined that the 'bonding' arrangements resulting from the contracts produced staff whose primary motivation was completing a contract, rather than an attraction to work in the organization. Automatic employment was therefore discontinued, requiring new staff to compete with nurses trained in other institutions for any available vacancies.

Staff unrest because of what was reported to be insensitive leadership, with difficult working conditions, demanded for and led to a radical leadership change. Institutional changes in 2017 ushered in a leadership that indicated a desire to engage with staff concerns. A hospital-wide culture survey carried out in mid-2017 led to the initiation of a culture change program, and subsequently, a refocusing of the institutional goals that created the beginnings of Westrum's generative culture [45], which in turn propagated a just culture by recognizing, educating, and actively replacing blame culture elements.

A non-punitive response to error is essential for the development of a patient safety culture. Such a culture encourages the reporting of medical errors. The institutional leadership must be willing to change systemic hurdles to the development of a wholesome patient safety culture that includes the abolition of archaic, paternalistic, and hierarchical cultures that frequently shield offenders from scrutiny and permit continued patient safety lapses [48, 49]. The differences in the mean percent positive dimension scores across most domains of HSOPS in 2019 (Table 2), are likely the result of cumulative efforts of the KH leadership to engage with all cadres of staff, and initiatives to both build a just culture, and replace the prevailing blame culture, recognizing that an institutional just culture is required in order to cultivate a safety culture, in which patient safety is prioritized by everyone. Staff perception of hospital management support for patient safety rose from a baseline of 62.7% to 80.8% in 2019 (p<0.0001), and improved staff working hours, and reception of staff suggestions, amongst others (Table 2) contributed to the perceived improvement in patient safety culture. We also hypothesized that the improved working environment contributed to a reduction in staff turnover in 2019. Increased and improved hospital administration employee engagement was enshrined in the

hospital's five-year strategic plan as one of the pillars, creating a hitherto unspoken, and unfelt employee 'value'.

From the results of this study, the authors hypothesize that the institutional working environment is an important determinant of patient safety perception by staff; a just culture readily endears an institutional safety culture. The staff-perceived improvements in patient safety in this study were not validated by patient perceptions. Nevertheless, some authors have found positive correlations between improved patient safety perception by staff and patient perceptions and experiences [50, 51]. From these results, we hypothesized that a secure working environment is important for a patient safety culture to thrive. The changes noted in the 'Years worked in hospital' suggest a decrease in staff turnover in 2019, as evidenced by the decrease in 'under 1 year' staff and increase in '1 to 5 years' staff category. Previously, high staff turnover was attributed to better remuneration elsewhere; this study suggests however, that, while staff remuneration is important (it improved marginally in 2019), an improved working environment may have provided the attraction for staff to stay on.

A number of reports from Africa suggest that systemic issues are the main cause of medical errors [37, 38, 40]. These authors work in environments similar to that found in KH; in contradistinction, Hickner et al. in a study in the USA, found that the number of days off, the presence of a hospital mission statement on patient safety, and participation in in-house patient safety workshops were key to promoting a good patient safety culture in their environment [47].

The freedom of staff to report near-misses and other incidents without fear is critical to the development of an institutional patient safety culture. The fact that staff would self-report an incident is invaluable, as it indicates an understanding of, or desire for a safety culture. Barriers to error reporting include both individual and systemic factors: ignorance, lack of an effective medical error reporting system, and managerial policies, among others [52–55]. Ramirez et al. reviewed the effectiveness of a hospital incident-reporting system on patient safety improvement and found a significant reduction in near-miss or adverse events observed. They also noted a significant correlation between the patient safety workshops and the number of reports per month [56]. Although there was an increase in the percentage of staff not filling an incident form from the baseline (p<0.05), suggesting an improved patient safety culture in the current study, there were still 8% of staff filling six or more incident reports throughout the study period (Table 3).

In response to staff comments and sentiments, the hospital management implemented corrective measures in 2019: those addressed are ***italicized*** in Table 4. The last HSOPS was conducted in March 2019; in September 2019, an Electronic Medical System (EMS) was installed. Although the EMS, posed a number of challenges, it has created a more transparent system for catching errors, especially medication and perioperative errors. The management also renovated the patient/visitor's washrooms complex to improve and increase the existing facilities. Furthermore, a Quality Improvement Department that now oversees IPC and other patient safety issues was established. The institutional changes reported in this institutional safety culture journey were not single isolated events, but rather cumulative, incremental changes that are a natural product of quality improvement projects, as each improvement highlights the benefits, while exposing hitherto unrecognized hurdles to the institutional safety culture.

## Conclusions

To our knowledge, this is the longest longitudinal institutional HSOPS data presented from a hospital in sub-Saharan Africa. Although the dynamics of a high staff turnover and a difficult working climate presented challenges to creating a patient safety culture, this study

demonstrates that patient safety can be successfully implemented and embraced in sub-Saharan Africa, and that with appropriate championing by staff and institutional leadership, hospitals can become safer environments for patients. A supportive working environment enhanced by a healthcare institutional just culture creates the ingredients for improved patient safety culture. We hope that this report spurs other regional hospitals into developing and championing patient safety cultures within their institutions.

## Study limitations

This study had a number of limitations:

The authors excluded some HSOPS composite scores–although scaling down the number of composites is permitted as described in the HSOPS users guide [6], our results may not be generalizable outside KH.

Further, although the results span a period of seven years, in 2017, there was a major leadership change, resulting from a crisis in which workers revolted against a toxic working environment; the improvements noted in 2019, when compared to the baseline may therefore be a reflection of an improved working environment, either largely or in part.

Future studies measuring patient safety should include patient perception of safety; indeed a patient perception domain addition to the HSOPS tool should be considered, to enable patient validation of hospital staff perceptions, as well as empower patients in impacting safety initiatives in hospitals.

Finally, because of the high staff turnover, inculcating a sustainable institutional culture of safety is a challenge; the survey results over the seven years do not cover the same population of healthcare workers.

## Supporting information

**S1 Data. Summated study data.**
(DOCX)

## Acknowledgments

The authors are grateful to the Infection Prevention and Control Committee and the hospital administration for support throughout the study period.

## Author Contributions

**Conceptualization:** Peter M. Nthumba, Caroline Mwangi, Moses Odhiambo.

**Data curation:** Peter M. Nthumba, Caroline Mwangi, Moses Odhiambo.

**Formal analysis:** Peter M. Nthumba, Moses Odhiambo.

**Investigation:** Peter M. Nthumba, Caroline Mwangi.

**Methodology:** Peter M. Nthumba, Caroline Mwangi.

**Project administration:** Peter M. Nthumba.

**Resources:** Peter M. Nthumba, Moses Odhiambo.

**Software:** Peter M. Nthumba.

**Supervision:** Peter M. Nthumba.

**Validation:** Peter M. Nthumba, Caroline Mwangi, Moses Odhiambo.

**Visualization:** Peter M. Nthumba, Caroline Mwangi, Moses Odhiambo.

**Writing – original draft:** Peter M. Nthumba, Caroline Mwangi, Moses Odhiambo.

**Writing – review & editing:** Peter M. Nthumba, Caroline Mwangi, Moses Odhiambo.

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
