## [Decision Letter · Decision Letter 0]

23 Aug 2023

PGPH-D-23-00990

Developing a sustainable safety culture within a high staff-turnover environment in a rural sub-Sahara Hospital: A case study of AIC Kijabe Hospital.

Dear Dr. Nthumba,

Thank you for submitting your manuscript to PLOS Global Public Health. After careful consideration, we feel that it has merit but does not fully meet PLOS Global Public Health’s publication criteria as it currently stands. Therefore, we invite you to submit a revised version of the manuscript that addresses the points raised during the review process.

Please note that we have only been able to secure a single reviewer to assess your manuscript. We are issuing a decision on your manuscript at this point to prevent further delays in the evaluation of your manuscript. Please be aware that the editor who handles your revised manuscript might find it necessary to invite additional reviewers to assess this work once the revised manuscript is submitted. However, we will aim to proceed on the basis of this single review if possible. 

We look forward to receiving your revised manuscript.

Kind regards,

Julia Robinson

Executive Editor

Journal Requirements:

1. We noticed you have some minor occurrence of overlapping text with the following previous publications, which needs to be addressed:

- https://journals.lww.com/md-journal/Fulltext/2018/09210/Effectiveness_and_limitations_of_an.112.aspx

- https://www.emerald.com/insight/content/doi/10.1108/IJHCQA-07-2012-0072/full/html

- https://pso.ahrq.gov/sites/default/files/wysiwyg/npsdpatient-safety-culture-brief.pdf

- https://www.afro.who.int/sites/default/files/2020-10/Ghana%20Annual%20report%202019.pdf

In your revision ensure you cite all your sources (including your own works), and quote or rephrase any duplicated text outside the methods section. Further consideration is dependent on these concerns being addressed.

2. In the online submission form, you indicated that "Data availability statement: Data are available upon reasonable request. Please contact the corresponding author to discuss the feasibility of obtaining access to a deidentified data set". All PLOS journals now require all data underlying the findings described in their manuscript to be freely available to other researchers, either 1. In a public repository, 2. Within the manuscript itself, or 3. Uploaded as supplementary information.

Additional Editor Comments (if provided):

Reviewers' comments:

Reviewer's Responses to Questions

**Comments to the Author**

1. Does this manuscript meet PLOS Global Public Health’s publication criteria? Is the manuscript technically sound, and do the data support the conclusions? The manuscript must describe methodologically and ethically rigorous research with conclusions that are appropriately drawn based on the data presented.

Reviewer #1: Partly

2. Has the statistical analysis been performed appropriately and rigorously?

Reviewer #1: I don't know

3. Have the authors made all data underlying the findings in their manuscript fully available (please refer to the Data Availability Statement at the start of the manuscript PDF file)?

Reviewer #1: Yes

4. Is the manuscript presented in an intelligible fashion and written in standard English?

Reviewer #1: Yes

5. Review Comments to the Author

Reviewer #1: Well informed/written and comprehensive overview of patient safety culture in Africa, melded with primary data from the local hospital culture surveys over a seven-year period. Somewhat unusual format, but perhaps it “works” here.

The literature review seems basic and perhaps not systematic but generally informative and relevant. I have not re-searched the topic to closely examine whether they have picked up all the relevant papers, but most of what they've included seems good and germane.

I question the title--whether it is exactly appropriate for the article contained here. The "high turnover" seems to come a bit out of the blue, although we know this is a very important patient safety issue and seems especially true in this hospital. (tho we have nothing to compare it to). And the word "developing" also doesn't seem to fit with this paper which does little to describe or study what initiatives are being taken to “develop” such a culture. On the contrary it is really a reporting of the survey responses over time, which as the authors point out, is a innovative contribution in itself. Thus t the authors shouldn't use the title to claim something there specific study did not actually do.

A related problem is that someone could read this as a promotional cheerleading piece that selectively pulls out areas where there is bona fide improvement and touts them. Since the authors were involved in both overseeing quality initiatives in the hospital as well as conducting the survey and crafting the language here, I think they should go through the paper with a I on softening any cheerleading and promotional commentaries. Perhaps there is even a conflict of interest that needs to be acknowledged in the discussion. Better to let the "facts" speak for themselves --and there are a number of impressive areas of improvement particularly in the 2019 survey.

This brings us to the other concern about the paper in general. The improvements and findings are a hodgepodge of some areas getting better, others getting worse or staying the same, and others going up and down over time. It is somewhat helpful that they are providing context to perhaps explain some of these results (e.g. revolt of the staff about leadership and working conditions), but I think the authors should again carefully go through the paper and minimize attributions to these changing numbers, certainly highlighting that these are at best hypotheses and associations rather than conclusions that can be drawn from the data.

Of course when we think about staff turnover and chaos in culture, the last three years are even more so, and would be interesting to see how that impacted on these metrics. Did the authors do a survey in 2021? Perhaps not and thus it wouldn't be fair to expect that they would if the hospital was in the throes of the COVID crisis.

The authors mention a relatively high response rate although the numbers of responses are only in the several hundred. Is this in fact the total number of staff in the surveyed categories? A staffing number of 650 is mentioned but it appears that the denominator for this study is something smaller. There seems to be an error/typo in the text when you discuss staff not generally in direct contact with patients – these groups of staff excluded? If so that needs to be stated.

The authors mention "six champions who ran the day-to-day activities of the project." What exactly is "the project." I suspect they are referring to the quality and safety unit of the hospital but these descriptors are not well enough defined here.

The qualitative comments thematic analysis seems a tad week. I would've hoped that the authors could've solicited more substantive and critical comments from those surveyed. Instead might this be related (for example) to the higher response rate where workers were required to fill this out but many did so perfunctory. The responses seem mostly short and more platitudes than critical insights.

Staff perceptions of hospital management support for patient safety rose from 62% to 80% – this and several other big jumps struck me as both impressive as well as potential red flags. Could there be unmeasured confounders or artifacts in what we are seeing here. These are obviously different employees (given the high turnover rate) but what about the mix of employees and the fact that this doesn't correlate with several other measures in the same survey.

“it is apparent from the study that the institutional working environment is an important determinant of patient safety perception by staff." I am not sure that the authors can justify concluding that there paper proves this. i.e. more of a correlation and a hypothesis, not necessarily something we can draw hard conclusions about here. .

The installation of an EMR must have been a huge confounder (and perhaps could be one factor to explain some of the dramatic changes in 2019).

In conclusions I would say it's the longest "longitudinal" institutional HS OPS data.

Figure 1 needs to be improved so the image is less blurry. I had trouble reading it.

Also for the editors, when I downloaded the PDF, the tables were not accessible when I clicked under supplemental material. Please fix this problem in editorial manager as I assume you want to make it easier for your reviewers. (Normally the PDF that is generated includes the tables from the paper).

6. PLOS authors have the option to publish the peer review history of their article (what does this mean?). If published, this will include your full peer review and any attached files.

**Do you want your identity to be public for this peer review?** For information about this choice, including consent withdrawal, please see our Privacy Policy.

Reviewer #1: No

---

## [Decision Letter · Decision Letter 1]

24 Apr 2024

PGPH-D-23-00990R1

Patient safety in a rural sub-Sahara Hospital: A 7-year experience at the AIC Kijabe Hospital, Kenya.

Dear Dr. Nthumba,

Thank you for submitting your manuscript to PLOS Global Public Health. After careful consideration, we feel that it has merit but does not fully meet PLOS Global Public Health’s publication criteria as it currently stands. Therefore, we invite you to submit a revised version of the manuscript that addresses the points raised during the review process.

We look forward to receiving your revised manuscript.

Kind regards,

Ari Natalia Probandari, PhD

Academic Editor

Journal Requirements:

1. Please amend your online Financial Disclosure statement. If you did not receive any funding for this study, please simply state: “The authors received no specific funding for this work.”

2. Please update your online Competing Interests statement. If you have no competing interests to declare, please state: “The authors have declared that no competing interests exist.”

3. We have noticed that you have uploaded Supporting Information files ("HSOPS DATA File.docx"), but you have not included a list of legends. Please add a full list of legends for your Supporting Information files after the references list.

Additional Editor Comments (if provided):

Please kindly response the remaining reviewers' comments point by point.

Reviewers' comments:

Reviewer's Responses to Questions

**Comments to the Author**

1. If the authors have adequately addressed your comments raised in a previous round of review and you feel that this manuscript is now acceptable for publication, you may indicate that here to bypass the “Comments to the Author” section, enter your conflict of interest statement in the “Confidential to Editor” section, and submit your "Accept" recommendation.

Reviewer #1: All comments have been addressed

Reviewer #2: All comments have been addressed

Reviewer #3: (No Response)

2. Does this manuscript meet PLOS Global Public Health’s publication criteria? Is the manuscript technically sound, and do the data support the conclusions? The manuscript must describe methodologically and ethically rigorous research with conclusions that are appropriately drawn based on the data presented.

Reviewer #1: Partly

Reviewer #2: Partly

Reviewer #3: Partly

3. Has the statistical analysis been performed appropriately and rigorously?

Reviewer #1: I don't know

Reviewer #2: I don't know

Reviewer #3: I don't know

4. Have the authors made all data underlying the findings in their manuscript fully available (please refer to the Data Availability Statement at the start of the manuscript PDF file)?

Reviewer #1: Yes

Reviewer #2: Yes

Reviewer #3: Yes

5. Is the manuscript presented in an intelligible fashion and written in standard English?

Reviewer #1: Yes

Reviewer #2: Yes

Reviewer #3: No

6. Review Comments to the Author

Reviewer #1: Mostly responsive revisions

No other changes at this point

Some of the prose could use a once over by a prose editor

Reviewer #2: Thankyou for giving me this opportunity to review the research article. It was my pleasure to think upon the innovative ways in which the article can be enhanced more profoundly.

The manuscript is a good in-depth study of patient safety culture at AIC Kijabe Hospital, providing unique insights into the evolution and challenges of building a safety culture in sub-Saharan African healthcare institutions.

Here are few minor changes that are required according to me:

ABSTRACT:

1. The Methods section lacks details on how the HSOPS tool was applied, including sample size, demographics, any modifications made to the tool to adapt it with the hospital needs.

2. The Results section needs a more specific information on magnitude of these improvements or any other trends observed over the study period.

3. Although staff turnover and leadership changes are acknowledged as problems, the conclusion retains that sub-Saharan Africa fulfils patient safety objectives. But rather than specific proof from the study's findings, it hinges on hypothesized relationships.

INTRODUCTION: A thorough description of patient safety culture, its barriers, and efforts to enhance it—particularly in sub-Saharan Africa—are given in the introduction. Nevertheless, a few crucial details are absent:

1. The specific objectives of the study are missing. The purpose of doing the study should be more emphasized upon.

2. There is no specific data of current patient safety culture in sub-Saharan Africa which may probably also serve as a baseline for the comparison of this study to the audience.

METHODS:

1. How were the surveys administered to the participants? Was there any modification done to the HSOPS tool?

2. Since the approach of the study design was narrative reviews, was there any potential bias in study selection and data synthesis?

3. The methods outline how hospital staff members are given the HSOPS tool after being given an overview of the tool and patient safety principles. Despite this, the methodology would be strengthened by providing information on the sampling strategy (such as convenience sampling or random sampling) and the steps taken to guarantee the sample's representativeness. Furthermore, even though using anonymous surveys aids in maintaining anonymity, it's very important to address any potential biases brought about by self-selection or non-response.

4. Also, the participant recruitment information is missing.

5. The section majorly lacks the data analysis part which tells about the coding process in qualitative data. If there were any theoretical frameworks used for it. For the quantitative data, there is no mention about the specific statistical tests used.

6. Transparency and adherence to ethical standards are enhanced by the Institutional Ethical Review Board's ethical approval being mentioned. But there is no justification provided for not obtaining the formal consent from participants. This particularly raises ethical concerns regarding informed consents and the participants’ rights.

RESULTS:

1. It would be helpful to include information on the distribution of respondents across different departments or units within the hospital to assess the representativeness of the sample.

2. Additional details on the methods used to calculate turnover rates and perceptions of support would strengthen the analysis.

3. Also, there can be an improvised detail on how was the missing data of 2013 was handled in the analysis to ensure validity of the results?

DISCUSSION:

1. The discussion emphasizes the necessity for validation through patient perceptions and experiences, accurately pointing out the limitations of depending only on staff opinions of patient safety culture. In subsequent research, it would be beneficial to talk about possible methods for integrating patient input into the evaluation of patient safety culture.

2. A more thorough examination of the contextual differences between healthcare settings in high-income countries and those in low- and middle-income countries, particularly in sub-Saharan Africa, and how these differences may influence patient safety culture, would be beneficial, even though the section compares the findings with those of studies conducted in the USA.

3. Furthermore, greater information on the precise steps taken and their effects on patient safety culture would be helpful, even if the discussion highlights the corrective measures that hospital management took in response to staff opinions and remarks.

CONCLUSION:

1. Since the study focused on a single institution, it would be beneficial to discuss the extent to which the findings may be applicable to the other healthcare settings in sub-Saharan Africa, considering variations in the organizational culture, resources and patient population.

2. Considering the study limitations, more details can be emphasized on how these limitations may have influenced the interpretation of the study findings or the generalizability of the results beyond AIC Kijabe Hospital.

Reviewer #3: Summary of the research and overall impression

The authors apply an adapted version of the Hospital Survey on Patient Safety Culture (HSOPS) as a tool to assess and compare institutional safety culture over a 7-year period (2013-2019) at Kijabe Hospital, Kenya. Given the perfunctory nature of safety and quality studies in Low- and Middle-Income Countries (LMICs), including Sub-Saharan African countries, I appreciate the efforts of the authors to document the introduction and institutionalization of a safety culture in their hospital and report on that over time from 2013-2019. The authors predominantly discuss the positive results from the 2019 survey in relation to the 2013 survey and discuss how contextual factors could explain some of the 2017 survey results. Discussing qualitative data about institutionalizing a safety culture could add considerable value to the study as institutionalizing a safety culture is difficult to be captured solely via a survey. It could be very valuable to know how safety measures improved and the hospital activities that that entailed between survey periods. It takes effort, activities, and communication to build and institutionalize a safety culture and the paper does not contribute to that process of construction and institutionalization of the safety culture over time. Finally, while the authors present data only from Kijabe Hospital in Kenya, and state that their “results may not be generalizable outside KH”, they claim that “this study demonstrates that patient safety ideals are embraced in sub-Sahara Africa.”

Discussion of specific areas for improvement

Major issues:

1) [page 10/53] The authors mention that: “In the week before the questionnaire was distributed, staff were taught the precepts of patient safety, and shown the African SUSP patient safety video [44].” I might have read this wrong, however paper eludes that this was done consistently across all 4 survey time points. Even so, this activity could be considered as priming the survey participants and could likely bias the survey results regarding safety culture.

2) [page 11/53] The authors mention that: “The team developed themes from the comments; these were then coded.” It is not very clear to follow how the qualitative analysis was done by the aforementioned statement. Usually, qualitative data is coded, and themes arise from the codes. It could be helpful if the authors expanded a little further on their data analysis approach.

3) [page 13/53] “Eleven major themes emerged from an analysis of the free comments from the 2019 survey (Table 3, Figure 1).” It is unclear whether the 2013, 2015, and 2017 surveys allowed participants to share their comments or whether only the 2019 survey included it and therefore the authors only report on qualitative aspects of the study for 2019. It could be helpful if the authors could clarify this aspect.

4) The authors state that “The initial survey helped establish the concepts of a patient safety culture and raise staff awareness about patient safety” and that “All the HSOPS results were reported to and discussed with senior management; the senior management addressed some of issues that came out of each report.” Given that conversations seem to have happened based on the aforementioned statements, this study could have been very informative and valuable with a richer qualitative aspect to help explain the reported safety variables over time. Institutionalizing a safety culture is difficult to be captured solely via a survey and survey results. Instead, it could be very valuable to know how safety measures improved and the hospital activities that that entailed between survey periods. It takes effort, activities, and communication to build and institutionalize a safety culture and the paper does not add to that process of construction and institutionalization of the safety culture over time. For example, looking at Table 2a, it would be very helpful to have more insights into how hospital management supported patient safety, how mistakes led to positive changes, how staff actively improved patient safety. Such findings could be valuable, generalizable and the lessons transferable to other settings for institutionalizing hospital safety culture. If such data had been collected but not reported in this study, I would strongly recommend including them in a subsequent draft.

5) The authors mention difficult working conditions that affected the 2017 results. While qualitative aspects between 2013-2015 or 2017-2019 could provide insights into building a culture of safety given the improved survey results, the 2015-2017 results could be powerful to gain insights into how difficult working conditions affect safety culture and safety outcomes via a more in-depth qualitative approach and further support the authors hypothesis about the relationship between just and safety cultures. If such data had been collected but not reported in this study, I would strongly recommend including them in a subsequent draft as well.

6) [page 12/53] It is important to survey different cadres as to their perceptions of safety and I appreciate the efforts of the authors to cover diverse cadres in the hospital. However, the wide range of cadres surveyed may dilute more pronounced results from cadres that are more directly involved with safety. Also, the cadre labelled as “others” seems to combine very different sub-cadres – as per the authors, the category “others” included “Infection Prevention and Control staff, medical records, housekeeping, and security personnel” – infection staff and security personnel could have very different perspectives of the safety culture and activities based on their proximity to patients and processes in the hospital.

7) The authors seem to be presenting contradicting statements and a clearer position could be helpful. The authors state that “this study demonstrates that patient safety ideals are embraced in sub-Sahara Africa” (pages 1, 5, 16) however, they are presenting data on only one hospital in Kenya, and the authors themselves state that “our results may not be generalizable outside KH” (page 16/53).

Minor/other issues:

8) The authors state that they “performed a narrative review of literature on patient safety in sub-Sahara Africa between January 2000 and December 2020.” The authors are providing the context and engaging with the literature to situate their study. I am not sure that this review of the literature needs to be framed as a “narrative review of the literature.”

9) The authors report that “the average response rate over the study period was 84.5%.” I am not sure how useful this is for the abstract of the paper. Perhaps the authors could consider providing a range for the response rate across the four different surveys over the 7-year period, like they do in the main body of the paper, instead of an average.

10) Table 2a is somewhat cumbersome to read - perhaps the results could be re-arranged so that the statistics appear after each column (2013-2015, 2015-2017, 2017-2019). In addition, the authors could indicate significant results with an asterisk or highlight the significant results on the table in an alternative way for better visibility.

11) As PLOS Global Health does not copyedit accepted manuscripts, I highly recommend going through the manuscript and correcting errors of that nature (e.g., use of acronyms without being explicitly defined, use of singular/plural (“chi-square tests was used”).

7. PLOS authors have the option to publish the peer review history of their article (what does this mean?). If published, this will include your full peer review and any attached files.

**Do you want your identity to be public for this peer review?** For information about this choice, including consent withdrawal, please see our Privacy Policy.

Reviewer #1: **Yes: **Gordon D Schiff

Reviewer #2: **Yes: **Dr. Nisha Mutalikdesai

Reviewer #3: No

---

## [Decision Letter · Decision Letter 2]

28 Aug 2024

PGPH-D-23-00990R2

Patient safety in a rural sub-Saharan Africa Hospital: A 7-year experience at the AIC Kijabe Hospital, Kenya.

Dear Dr. Nthumba,

Thank you for submitting your manuscript to PLOS Global Public Health. After careful consideration, we feel that it has merit but does not fully meet PLOS Global Public Health’s publication criteria as it currently stands. Therefore, we invite you to submit a revised version of the manuscript that addresses the points raised during the review process.

We look forward to receiving your revised manuscript.

Kind regards,

Connie Gan

Academic Editor

Journal Requirements:

1. We noticed you have some minor occurrence of overlapping text with the following previous publications, which needs to be addressed:

- https://journals.lww.com/md-journal/Fulltext/2018/09210/Effectiveness_and_limitations_of_an.112.aspx

- https://www.emerald.com/insight/content/doi/10.1108/IJHCQA-07-2012-0072/full/html

- https://pso.ahrq.gov/sites/default/files/wysiwyg/npsdpatient-safety-culture-brief.pdf

- https://www.afro.who.int/sites/default/files/2020-10/Ghana%20Annual%20report%202019.pdf

In your revision ensure you cite all your sources (including your own works), and quote or rephrase any duplicated text outside the methods section. Further consideration is dependent on these concerns being addressed.

Reviewers' comments:

Reviewer's Responses to Questions

**Comments to the Author**

1. If the authors have adequately addressed your comments raised in a previous round of review and you feel that this manuscript is now acceptable for publication, you may indicate that here to bypass the “Comments to the Author” section, enter your conflict of interest statement in the “Confidential to Editor” section, and submit your "Accept" recommendation.

Reviewer #2: All comments have been addressed

Reviewer #3: (No Response)

2. Does this manuscript meet PLOS Global Public Health’s publication criteria? Is the manuscript technically sound, and do the data support the conclusions? The manuscript must describe methodologically and ethically rigorous research with conclusions that are appropriately drawn based on the data presented.

Reviewer #2: Yes

Reviewer #3: No

3. Has the statistical analysis been performed appropriately and rigorously?

Reviewer #2: I don't know

Reviewer #3: I don't know

4. Have the authors made all data underlying the findings in their manuscript fully available (please refer to the Data Availability Statement at the start of the manuscript PDF file)?

Reviewer #2: Yes

Reviewer #3: No

5. Is the manuscript presented in an intelligible fashion and written in standard English?

Reviewer #2: Yes

Reviewer #3: No

6. Review Comments to the Author

Reviewer #2: No comments.

Reviewer #3: Dear Authors,

Thank you very much again for your hard work, continuous efforts for improvement of your manuscript, and for providing clarifications on the issues I raised in my previous review. KH could be considered a unique case regarding institutionalizing safety culture and with some re-framing, your paper could make that case more insightful, as well as more generalizable. Below are some salient aspects that I believe merit your attention further.

1) 2013/2015/2017 “Free” comments

Thank you for the clarity regarding the lack of “free” comments in years 2013, 2015 and 2017. If I am understanding correctly, this means that the changes which are now listed in Table 3 as “Changes effected” reflect changes made at KH after the results of the 2019 HSOPS survey that you are reporting. Even though helpful to know, these could help explain the results of a future 2021 survey, but I am not sure how the added column is helpful in explaining the safety culture trends between 2013-2019, which is the aim of the paper.

2) 2019 “Free” comments

I understand that no qualitative data were not collected in 2013/2015/2017. I wonder how much data was collected in 2019 to necessitate analyzing in NVivo? Would it be possible to make that data available or talk about that data analysis a little more?

3) Safety Culture interventions & 2013/2015/2017 “Free” comments

Since there is no qualitative data from 2013-2017, it would be helpful if you could tell us more about the changes made at the organizational level to institutionalize safety culture from 2013-2019. While these might not have come from the “free” survey comments, I believe it is necessary to show us what you did as a hospital to explain improved results you are reporting. As it is right now, this paper is not guiding us as to what you did to institutionalize safety culture from 2013-2019 and how the positive results that you report through the HSOPS tool matter or serve as lessons learned for institutionalizing safety culture.

On page 6/70, you mention: “Patient safety champions, a generative institutional leadership that is supportive of patient safety, are important for the development of an institutional safety culture.” On page 17/70, at the end of the Discussion and just before the Conclusions of the paper, you mention that “Furthermore, a Quality Improvement Department that now oversees IPC and other patient safety issues was established.” This is very important information, and such information should not be presented so briefly and/or at the very end of the paper. For example, as a reader and a reviewer, I would like to know when was this Quality Improvement Department put in place? How many employees did the department have and with what expertise or training around safety? What were some of the activities of this department? What activities, if any, were introduced subsequently to support/complement the initial activities of the department? Is this department the only structure that was put in place from 2013-2019 to support safety culture or were other processes and structures subsequently introduced to support safety culture at KH?

Perhaps it could be helpful to think about a timeline schematic of the different structures and processes that were put in place in KH from 2013-2019 and that could help explain the HSOPS results. While this relationship is not causal, it could help the reader to better understand what you did as a hospital to establish safety culture. If all structures, processes, initiatives, activities surrounding safety culture were to be presented in a timeline from 2013-2019, that would be very helpful to see how some potential changes contributed to the positive survey results that you are basing your paper on. This could tell us a lot more about your difficult and enriching journey in institutionalizing safety culture and could help other hospitals in their own safety culture journeys as well.

I believe that your focus has been predominantly on the HSOPS tool – I would like to kindly ask you to take a step back from the paper that you have worked so hard and diligently on and reflect on what the results of the survey could mean to a hospital administrator that would like to follow your safety culture practices and what you accomplished if the administrator does not know what you did at KH to achieve positive safety culture results?

I believe this would be a strong paper if you could show that you introduced x interventions, you have all the contextual organizational and leadership challenges, and despite it all, you have a change in safety culture that you can show using the HSOPS tool – a combination of these three aspects would strengthen your case about the positive results you see in 2019. I am sure you have put a lot of effort in this initiative; you could bring all that work to the foreground, shift the narrative, and tell us what you did in more detail and your rich experience instead of the HSOPS survey dominating the paper.

4) Literature review

The authors state that they “reviewed literature on patient safety in sub-Sahara Africa between January 2000 and December 2020.” Similar to my comment in the previous review, the authors are providing the context and engaging with the literature to situate their study. I am still not sure that this review of the literature needs to be stated as such in the paper.

5) “Just culture” and “Safety culture”:

In the abstract and at the end of the introduction you state: “Creating an institutional just culture creates a patient safety culture.” I am sorry if I am missing something in the manuscript, but I cannot really identify other places in the paper where you mention “just culture” and its association to safety culture.

6) Positive Dimension Score

The main outcome across the surveys is the percent positive dimension score. Could you please provide more information how that is calculated?

7) Table 2b and Turnover

If one looks carefully at Table 2b, there are groups that I am not sure how to interpret:

- Are there are really employees that work 100 hours or more hours?

- It is not immediately obvious or stated if you are calculating a median or an average for the turnover at the 2015-2017-2019. If you are considering all the different groups equally, that means that you are giving equal weight in knowledge and experience to the employee working for < 20 hrs per week and an employee who is working 40-59 hours. Could you look at and discuss the turnover across the group that works the median number of hours of work at KH (40-59 hours)? If we only look at that group, the turnover oscillates between 2015-2017-2019.

- As a reader/reviewer, I am wondering of the different explanatory options possible: Did increased safety culture improve turnover because employees felt they worked in a safer environment and therefore wanted to stay working at KH? An alternative explanation (and somewhat contradictory) could be that safety culture exists despite high turnover; in that case, high turnover makes the case even stronger for KH; it could mean that the institutional measures and structures set in place to institutionalize safety culture were able to overcome a crucial issue such as employee turnover and despite the turnover, the 2019 results are still positive than 2013. I do not know what to consider as the paper in its current form does not discuss any insights from your depth of the experience. Your voice and perspective is powerful and we do not hear it through this paper in its current form.

- I am not sure I understand why “Mistakes caught and corrected before affecting patients,” “We discuss ways to prevent errors from happening again,” and “How many events reports have you filled?” are under “Total Hospital Staff Turnover.” [bolded]

- Was turnover measured every 2 years when the survey was administered?

8) Overall Copy Editing

I would kindly encourage an extensive copy editing to the paper. Some examples of aspects that still need corrections include the following:

- Sub-Saharan African Hospital

- Author et al.: some contain ., some do not

- While the grand majority of references are listed by [Number], some references are in the text

- The language is somewhat informalat different points, i.e. “Mallouli et al from Tunisia used a French tool...”; this could be rephrased: “The use of HSOPS in Africa has been reported in a few studies [37-40], while Mallouli et al. also employed an alternative tool to assess patient safety in primary healthcare in Tunisia [41].”

- The inconsistent use of capitalized words in the middle of sentences: “…safety culture; stress and fatigue; inadequate training, education, and manpower issues; Lack of appropriate knowledge and availability of knowledge, transfer of knowledge; Errors in the structure and process of care; Adverse events...”

Please take a thorough look at the manuscript in its entirety for occurrences like the aforementioned ones for punctuation and prose.

---

## [Editor Report · Decision Letter 3]

24 Sep 2024

PGPH-D-23-00990R3

Patient safety in a rural sub-Saharan Africa hospital: A 7-year experience at the AIC Kijabe Hospital, Kenya.

Dear Dr. Nthumba,

Thank you for submitting your manuscript to PLOS Global Public Health. After careful consideration, we feel that it has merit but does not fully meet PLOS Global Public Health’s publication criteria as it currently stands. Therefore, we invite you to submit a revised version of the manuscript that addresses the points raised during the review process.

We look forward to receiving your revised manuscript.

Kind regards,

Connie Cai Ru Gan

Academic Editor
---

## [Editor Report · Decision Letter 4]

22 Oct 2024

Patient safety in a rural sub-Saharan Africa hospital: A 7-year experience at the AIC Kijabe Hospital, Kenya.

PGPH-D-23-00990R4

Dear Dr. Nthumba,

We are pleased to inform you that your manuscript 'Patient safety in a rural sub-Saharan Africa hospital: A 7-year experience at the AIC Kijabe Hospital, Kenya.' has been provisionally accepted for publication in PLOS Global Public Health.

Best regards,

Connie Cai Ru Gan

Academic Editor
